# Analysis of Acute and Chronic Methamphetamine Treatment in Mice on Gdnf System Expression Reveals a Potential Mechanism of Schizophrenia Susceptibility

**DOI:** 10.3390/biom13091428

**Published:** 2023-09-21

**Authors:** Laoise Casserly, Daniel R. Garton, Ana Montaño-Rodriguez, Jaan-Olle Andressoo

**Affiliations:** 1Department of Pharmacology, Faculty of Medicine, Helsinki Institute of Life Science, University of Helsinki, 00290 Helsinki, Finland; 2Division of Neurogeriatrics, Department of Neurobiology, Care Science and Society (NVS), Karolinska Institutet, 17177 Stockholm, Sweden

**Keywords:** schizophrenia, dopamine, GDNF, GFRa1, RET, monoamines, serotonin, psychosis, methamphetamine

## Abstract

The increase in presynaptic striatal dopamine is the main dopaminergic abnormality in schizophrenia (SCZ). SCZ is primarily treated by modulating the activity of monoamine systems, with a focus on dopamine and serotonin receptors. Glial cell line-derived neurotrophic factor (GDNF) is a strong dopaminergic factor, that recently was shown to correlate with SCZ in human CSF and in striatal tissue. A 2-3-fold increase in GDNF in the brain was sufficient to induce SCZ-like dopaminergic and behavioural changes in mice. Here, we analysed the effect of acute, chronic, and embryonic methamphetamine, a drug known to enhance the risk of psychosis, on *Gdnf* and its receptors, *Gfra1* and *Ret*, as well as on monoamine metabolism-related gene expression in the mouse brain. We found that acute methamphetamine application increases *Gdnf* expression in the striatum and chronic methamphetamine decreases the striatal expression of GDNF receptors *Gfra1* and *Ret*. Both chronic and acute methamphetamine treatment upregulated the expression of genes related to dopamine and serotonin metabolism in the striatum, prefrontal cortex, and substantia nigra. Our results suggest a potential mechanism as to how methamphetamine elicits individual psychosis risk in young adults—variation in initial striatal GDNF induction and subsequent GFRα1 and RET downregulation may determine individual susceptibility to psychosis. Our results may guide future experiments and precision medicine development for methamphetamine-induced psychosis using GDNF/GFRa1/RET antagonists.

## 1. Introduction

Schizophrenia (SCZ) is a debilitating psychiatric disorder with remarkable interindividual variation. SCZ has many hypotheses surrounding its pathophysiology, including theories relating to altered GABA-ergic and glutamatergic systems, inflammation, and oxidative stress. However, the molecular drivers of the disease remain relatively poorly understood. The most consistent and scientifically supported hypothesis describes altered brain dopaminergic function as a driver and/or contributor to SCZ [1,2]. This hypothesis is supported by multiple independent lines of evidence. A serendipitous discovery in the early 1950s ′s revealed that dopamine (DA) receptor 2 (D2R) blocking drugs can reverse psychosis, whereas DA-enhancing drugs, like methamphetamine, enhance the disease and increase psychosis risk [3,4,5]. In line with this, one of the more robust abnormalities in schizophrenia is an increase in striatal dopamine activity. Brain imaging studies, using SPECT and PET, reveal increased striatal DA synthesis and/or release as a consistent, but non-ubiquitous, feature in individuals with SCZ and individuals with increased risk for SCZ [6,7,8]. More recently, it was shown that about half of first-episode patients (FEP) display enhanced striatal dopamine metabolism and that only those patients with elevated striatal dopamine synthesis respond to treatment with antipsychotics [9], which notably all target the D2R [7,8,10].

The fact that only about half of individuals with SCZ respond to anti-psychotics and that there are no single causal gene mutations for SCZ, collectively imply that a one-size-fits-all explanation for SCZ does not exist. Hence, to create precision medicine in the future, targeting enhanced striatal dopamine function in SCZ, we first need to understand the mechanisms driving the increased striatal dopamine and altered monoamine metabolism in SCZ [1,2]. This information can then be used to define disease subsets, which may eventually lead to the development of subgroup-specific precision medicine.

We have recently made progress in this direction. GDNF is a strong DA neuron function-enhancing factor. We have found that GDNF protein levels in the CSF positively correlate with disease severity in FEP SCZ patients [2]. Furthermore, we found that GDNF mRNA levels are increased in the post-mortem striatum of SCZ patients and that a similar increase in GDNF expression in mice is sufficient to trigger an SCZ-like increase in striatal dopamine, in dopaminergic system function, gene expression, neurophysiology, and animal behavior [2].

The striatum is a key site of dopamine metabolism in the brain. GDNF is expressed specifically in the parvalbumin-expressing interneurons of the striatum, which regulate striatal output [11,12]. Previously, we found that striatal GDNF levels regulate amphetamine-induced DA release, and DA transporter reversal time, in striatal dopamine axons [13] and that endogenous GDNF levels regulate striatal DA release and reuptake [14]. Furthermore, it is well established that excess ectopically applied GDNF strongly stimulates striatal dopaminergic function when GDNF is delivered into Parkinson’s disease models, where striatal dopamine fibers degenerate [15,16,17,18,19,20,21].

Methamphetamine strongly enhances striatal DA metabolism and induces psychosis in some individuals, with the highest prevalence in adolescence [5,22,23,24,25]. Methamphetamine is often used to replicate schizophrenia-like symptoms in animals [26,27,28,29,30,31]. However, mechanisms related to methamphetamine-induced susceptibility to psychosis, and why some individuals are more susceptible than others, have remained unknown. Here, we performed an exploratory study to analyze if, and how, methamphetamine modulates spatiotemporal gene expression of *Gdnf*, and its receptors *Gfrα1* and *Ret*, in the brain in acute, chronic, and embryonic regimens.

We found that acute methamphetamine induces striatal *Gdnf* mRNA expression, whereas chronic methamphetamine induces downregulation of striatal GDNF receptors *Gfrα1* and *Ret* mRNA expression levels. This result suggests a possible mechanism for individual differences in psychosis induction—individuals with a greater acute striatal *GDNF* induction and a lower *GFRα1* and *RET* downregulation upon chronic drug application are likely the most susceptible to psychosis. Methamphetamine also upregulated the expression of the majority of dopamine and serotonin metabolism-related genes in the prefrontal cortex (PFC), striatum, and substantia nigra (SN), supporting the monoamine hypotheses of SCZ (Figure 1).

## 2. Materials and Methods

### 2.1. Animals

Animal experiments were conducted according to the 3R principles of the European Union Directive 2010/63/EU governing the care and use of experimental animals. Experiments were also conducted according to local laws and regulations (Finnish Act on the Protection of Animals Used for Scientific or Educational Purposes (491/2013), Government Decree on the Protection of Animals Used for Scientific or Educational Purposes (564/2013)). The protocols were authorized by the National Animal Experiment Board of Finland (ESAVI/12046/04.10.07/2017—Osahanke 1). Mice were maintained in a 129Ola/ICR/C57bl6 mixed genetic background. In order to replicate a similar genetic background used in Mätlik et al. [2], mice were maintained in a mixed 129Ola/ICR/C57bl6 background where a 2-to-4-fold midgestational upregulation of GDNF resulted in a spectrum of SCZ-like features from neurophysiology to animal behaviour. Mice were group-housed with free food and water access under a 12 h light-dark cycle and room temperature of 21 ± 1 °C. Bedding (aspen chips, Tapvei, Harjumaa, Estonia) and nest material (Tapvei, Harjumaa, Estonia) were changed weekly. Researchers were blinded to mouse genotypes during tissue collection and processing. Male mice were used for acute and chronic methamphetamine experiments and female dams were used for embryonic methamphetamine experiments.

### 2.2. Methamphetamine Injections

To assess changes in mRNA levels in response to methamphetamine, wild-type (WT) adolescent (1 month old) male mice were treated with methamphetamine following either an acute or chronic dosing regimen [32] (Figure 2). For acute methamphetamine (aMETH) treatment, mice were given IP injections of escalating doses of methamphetamine, diluted in 0.9% saline, over three days (1 mg/kg on day one, 2 mg/kg on day two, 4 mg/kg on day three). For chronic methamphetamine treatment (cMETH), mice were given IP injections of escalating doses of methamphetamine diluted in 0.9% saline for three weeks. Escalating doses were used to replicate methamphetamine abuse patterns observed in humans, where humans normally use lower doses of methamphetamine initially before gradually increasing the dose [33,34,35]. In week one, mice received 1 mg/kg methamphetamine once daily for five days. In week two, mice received 2 mg/kg methamphetamine twice daily for five days. In week three, mice received 4 mg/kg methamphetamine twice daily for five days. The two-day rest period between the five days of methamphetamine dosing was used in an attempt to replicate a temporary cessation of the drug intake in humans associated with psychostimulant withdrawal [35,36]. Similar dosing regimens have been used successfully in previous studies [34,37]. Wild-type mice injected only with saline were used as controls for both regimens. Mice were sacrificed by cervical dislocation and decapitation and dissected approximately 1–2 h post-final injection for acute regimens and 24 h post-final injection for chronic regimens. Brains were removed, placed into a chilled cutting block on ice, and quickly dissected according to the areas of interest and immediately frozen on dry ice and stored at −80 °C. To measure changes in mRNA levels in embryos, pregnant dams were injected IP with 1 mg/kg methamphetamine diluted in 0.9% saline solution once daily for two days (E11–E12; Figure 2). Saline-injected wild-type mice were used as controls. The pregnant dams were sacrificed 1–2 h post-final injection and the embryos (E12.5) were dissected on ice immediately. All tissues were snap-frozen on dry ice and stored at −80 °C until mRNA analysis.

### 2.3. Reverse Transcription and Quantitative PCR

RNA was isolated using a Trizol reagent (Thermo Fischer Scientific, Vantaa, Finland). 400 ng of total RNA per sample was treated with RNase-free DNase I (Thermo Fischer Scientific, Vantaa, Finland). DNase I was inactivated with 50 mM EDTA at 65 °C for 10 min. The reverse transcription reaction was performed using random hexamer primers and RevertAid Reverse Transcriptase (Thermo Fischer Scientific, Vantaa, Finland). Complementary DNA (cDNA) was diluted 1:10 and stored at 20 °C until used for qPCR.

Quantitative PCR was performed using LightCycler 480 SYBER Green I Master (Roche, Espoo, Finland) and 100 μM primers, diluted 1:20, in 384-well plates with 10 μL total volume. All samples were analysed in triplicates or duplicates. Each reaction contained two negative controls (water and minus-reverse transcription). A combination of B-Actin, Ribosomal Protein S6, Beta-2-Microglobulin, and Gapdh were used as housekeepers for normalisation. Primer sequences are provided in Table 1.

### 2.4. Statistical Analysis

All results are presented as the mean ± SEM. Normality was first assessed in all data sets to ensure normal distributions, and outliers were removed justified by Grubbs’ test. To then assess nominal significance, a Welch’s *t*-test was performed to compare specific differences due to methamphetamine treatment between mRNA expression levels of specific genes. However, as this is subject to false positives and false negatives, to simultaneously limit errors in comparing variance both across genes and between treatments, we performed a two-way ANOVA followed by the Holm-Sidak correction for multiple comparisons.

Statistical analysis was performed using GraphPad Prism 8.4.2. Quantitative PCR data was analysed as previously described [38] using the geometric mean of reference genes for normalisation. The statistical significance level was set at *p* < 0.05.

## 3. Results

### 3.1. Acute Methamphetamine Application Induces a 2-Fold Increase in Striatal Gdnf Expression

A recent study revealed that a 2–3-fold increase in endogenous GDNF expression in the adult striatum and the embryonic brain induces dopaminergic changes similar to that seen in schizophrenia [2]. Methamphetamine, on the other hand, induces psychosis with incomplete penetrance [22]. Here, we investigated the effect of acute methamphetamine application (Figure 2) on the mRNA expression of *Gdnf* and its receptors *Gfrα1* and *Ret*, as well as on the expression of a range of marker genes covering dopaminergic and serotonergic systems, as selected from gene data sets from GeneOntology (GO:0001963, GO:0006837, GO:0035860) [39,40] in the striatum, prefrontal cortex and substantia nigra. Since parvalbumin (*Pvalb*) expressing GABA-ergic interneurons are found to be responsible for the majority of GDNF produced in the rodent striatum [11,12], we also assessed *Pvalb* mRNA levels, which may provide further insight into the signalling capacity of GDNF in response to methamphetamine. Our results show that an acute methamphetamine treatment induces about a 2-fold increase in *Gdnf* mRNA in the striatum (*p* < 0.05 *, *p* < 0.001 ###, Figure 3B). However, acute methamphetamine treatment had no effect on the mRNA levels of *Gfra1*, *Ret*, and *Pvalb* in the striatum, prefrontal cortex, and substantia nigra (Figure 3).

### 3.2. Acute Methamphetamine Treatment Enhances Expression of Genes Related to Dopaminergic and Serotonergic System Function

The dopamine hypothesis of schizophrenia theorizes that hyperactivity of dopamine 2 receptor (D2R) neurotransmission in subcortical and limbic regions of the brain may contribute to positive symptoms of schizophrenia, potentially in part due to an increase in D2R density [41,42,43,44,45]. Negative and cognitive symptoms of schizophrenia on the other hand are believed to be associated, at least in part, with cortical hypofunctionality of the dopamine 1 receptor [46,47]. We observed a trend toward increased *D2R* mRNA levels in response to acute methamphetamine treatment in the prefrontal cortex (*p* = 0.051, Figure 3A). No changes were observed in *D1Ra* mRNA levels in the mouse brain in response to acute methamphetamine (Figure 3).

Tyrosine hydroxylase (TH) is the rate-limiting enzyme in the production of dopamine [48]. Post-mortem analysis in SCZ patients yielded inconsistent results, with some studies showing increased tyrosine hydroxylase levels in the substantia nigra [49], while others find no difference [50,51]. We found that acute methamphetamine treatment induces a statistically significant increase in *TH* mRNA expression in the prefrontal cortex (*p* < 0.01 **, Figure 3A).

Polymorphisms in the serotonin transporter gene have been linked to treatment-resistant schizophrenia [52] and changes in *SERT* expression in brain tissue from individuals with schizophrenia have been documented [53]. We found that acute methamphetamine application increases *Sert* mRNA expression in the substantia nigra (*p* < 0.0001 ***, Figure 3C), with no changes observed in the striatum or prefrontal cortex.

Taken together, our results show that an acute methamphetamine regimen enhances the expression of key metabolic genes in both dopaminergic and serotonergic systems.

### 3.3. Chronic Methamphetamine Treatment Upregulates Ret Expression in the Substantia Nigra but Downregulates Gfra1 and Ret mRNA Levels in the Striatum

To compare the effects of chronic methamphetamine treatment with acute methamphetamine treatment, we next used a chronic regimen with 1–2 daily methamphetamine injections for three weeks (Figure 2) and analysed the same set of genes in the same brain regions. We found that *Ret* mRNA expression is significantly increased in the substantia nigra in response to chronic methamphetamine treatment (*p* < 0.05 *, Figure 4C). Furthermore, we found a significant reduction in both *Gfra1* and *Ret* mRNA levels in the striatum (*p* < 0.05 *, Figure 4B) and we found that chronic methamphetamine nonsignificantly enhances the expression of *Pvalb*, which marks *Gdnf* expressing interneurons [11] in the striatum (*p* = 0.07, Figure 4B).

### 3.4. Chronic Methamphetamine Maintain Elevated D2R Expression in the Prefrontal Cortex and Induces DAT Expression in the Substantia Nigra

Somewhat surprisingly, we found that despite the initial boost in *SERT* and *TH* encoding mRNA expression observed post-acute methamphetamine application, an increase in their expression is no longer observed upon chronic treatment. However, increased *D2R* expression in the PFC was observed in both acute and chronic treatments (Figure 3A and Figure 4A). We also found that chronic methamphetamine treatment increases *DAT* expression in the substantia nigra (Figure 4). Serotonergic markers were largely unchanged post-chronic methamphetamine treatment, except for a significant reduction in *Htr2b* mRNA in the substantia nigra (*p* < 0.05 #, Figure 4C).

### 3.5. The Effect of Acute E11–E12.5 Methamphetamine Application on GDNF and Monoamine Systems-Related Gene Expression in the Brain

Schizophrenia has been associated with complications in pregnancy, including prenatal exposure to influenza, prenatal nutritional deprivation, and rhesus incompatibility [54,55,56,57]. A recently published study suggests that an increase in GDNF expression in the CNS, at around midgestation (E12.5), results in schizophrenia-like changes in dopamine metabolism and animal behaviour. Therefore, we analysed whether acute methamphetamine exposure during E11-E12 (Figure 2) could induce *Gdnf* expression at E12.5 in the whole brain. We analysed *Gdnf* induction during pregnancy since midgestational upregulation of GDNF induced a range of SCZ-like features in mice [2]. Thus, we explored whether methamphetamine usage during pregnancy induces a *Gdnf* peak in the embryonal brain as a possible risk factor for SCZ later in life.

We also analysed the effect of methamphetamine on the expression of the above set of monoamine metabolism-related genes. We found that acute methamphetamine induces *D1R* and adenosine receptor Adora2a (*A2A*) encoding mRNA expression in the embryonic brain at E12.5 (Figure 5). However, acute methamphetamine treatment has no effect on the expression of *Gdnf* and its receptors at this age, nor did we find any changes in the monoamine metabolism-related gene expression that were altered in adolescent mice upon acute and chronic methamphetamine exposure at that age.

## 4. Discussion

A presynaptic increase in striatal dopamine is the main dopaminergic anomaly in schizophrenia [6,7,8]. GDNF is specifically expressed in the striatum in parvalbumin-expressing interneurons [11,12], which regulate striatal output [58]. In the striatum, GDNF receptor RET is exclusively expressed in dopamine axons. These axons project to the striatum from the substantia nigra, where the cell bodies of dopamine neurons are located [15] (Figure 6). GDNF strongly enhances the expression of its receptors, Gfrα1 and Ret, as shown by others [59,60] and by us, in various brain regions, including the substantia nigra [2,14]. Importantly, GDNF is one of the strongest factors known for enhancing dopamine neuron metabolism and function [2,61,62,63] and GDNF elicits its dopamine-enhancing effects on dopamine neurons via GFRα1 and RET receptors [64,65]. However, the role of the GDNF system in striatal hyperdopaminergia observed in schizophrenia has remained obscure. There are several reasons for this.

First, it is currently unknown how GDNF levels in serum and CSF reflect GDNF levels in the brain. This, combined with a well-known lack of high-affinity antibodies that exclusively recognize endogenous GDNF, has complicated the drawing of a firm conclusion on GDNF biology in schizophrenia. Reflecting this, several studies report contradicting results in patients, with some claiming no change in GDNF serum levels [66,67,68], while other studies report decreased GDNF [69,70], with one of the aforementioned studies finding a correlation between increased GDNF and cognitive and attentional deficits in schizophrenia [68]. Our previous analysis using a hyper-sensitive endogenous GDNF protein detection system, available at Olink Ltd., Uppsala, Sweden, showed that GDNF levels correlate with schizophrenia in the CSF and that *GDNF* mRNA levels are increased in the striatum in schizophrenia patients [2]. Importantly, in mice, we established that a similar increase in GDNF results in schizophrenia-like molecular, neurophysiological, and behavioral changes, with increased presynaptic dopamine levels and release [2]. In our previous work, we therefore demonstrated that a 2- to 4-fold increase of GDNF from its endogenous locus both during midgestation and adulthood is sufficient to trigger various SCZ-like features in mice [2]. Methamphetamine, on the other hand, is well known to increase the risk of psychosis in humans [22,71] and has been used to model SCZ in animals [32,34,37,72]. Mice given the dosing regimen we have selected have been shown to exhibit greater amphetamine-induced locomotion indicating schizophrenia-like striatal hyperdopaminergia, and schizophrenia-like changes to the brain proteome as well as alterations in paired-pulse inhibition in mice with mutations related to schizophrenia susceptibility. [32,34,37,72]. Hence, here we performed an exploratory study to see if and how methamphetamine, applied at the doses and application regimen used in the above studies to induce SCZ-like behavior in animals, induces and modulates the expression of genes related to the GDNF system. As multiple studies reveal that *Gdnf* mRNA levels correlate with GDNF protein levels in the mouse brain [2,14,73,74], in this current exploratory study, we decided to measure *Gdnf* mRNA levels. Additionally, previous studies have demonstrated that *Gfra1* and *Ret* mRNA levels also correlate with protein levels in mice [75,76,77,78].

Here, our analysis of the acute and chronic effects of methamphetamine, a known risk enhancer for psychosis, gave further insight into how GDNF expression may relate to susceptibility to schizophrenia. We show that, in mice, acute methamphetamine treatment induces about a 2-fold increase in striatal *Gdnf* expression. The chronic methamphetamine regimen, on the other hand, induces a reduction in *Ret* and *Gfra1* encoding mRNA expression in the striatum. This is important, as the observed dual effect suggests a mechanism as to how methamphetamine induces psychosis with incomplete penetrance—individuals with higher GDNF induction and less prominent GFRa1/RET downregulation are more likely to develop psychosis. This, when supported by further analysis in methamphetamine-induced psychosis patients, suggests that temporary inhibition of GDNF signaling may serve as a future precision medicine for the treatment of methamphetamine-induced psychosis patients with elevated GDNF responses. This could be implemented using transient treatment with already available RET inhibitors Selpercatinib and Pralsetinib, which do reach the brain [79,80,81]. Selpercatinib and Pralsetinib are recently developed drugs, used to treat specific types of cancer [79,80,81]. Or this could be implemented using another yet-to-be-developed drug targeting, for example, GDNF-GFRa1 interactions. However, in the absence of data on RET phosphorylation and direct demonstration of reduced RET activity, this hypothesis, at this point, remains speculative. Nevertheless, this potential mechanism can guide future studies towards a novel avenue for psychopharmacological development of SCZ treatments for a subset of patients. Importantly, GDNF is well known to enhance striatal DA [2,14,82], and striatal hyperdopaminergia is strongly linked to SCZ [1,2,8,9]. Our results reveal a correlation between methamphetamine and GDNF induction, suggesting mechanistically that potentially methamphetamine could predispose one to or increase one’s risk of developing SCZ via GDNF upregulation-inducing STR hyperdopaminergia.

Furthermore, our analysis suggests that *RET* mRNA axonal localization is an actively controlled process, which is regulated by methamphetamine application. Upon chronic methamphetamine application, we observe an increase in *Ret* mRNA expression in the substantia nigra, with a simultaneous decrease in *Ret* mRNA expression in the striatum (Figure 4). Since *Ret* is only expressed in dopamine neurons and their axonal projections to the striatum [83] are the main source of striatal *Ret* mRNA [84], the most likely explanation is the existence of an actively controlled soma-to-axon *Ret* mRNA transportation mechanism which is modulated by methamphetamine. Future analysis of *Ret* mRNA localization would be useful for supporting this hypothesis.

Methamphetamine use among pregnant women is increasing, and substance abuse during pregnancy is associated with an increased risk of neurodevelopmental disorders [85]. However, drug-dependent mothers may use multiple drugs with various pharmacological properties, making it hard to determine the effect of a single drug on embryonic development. As we have already shown that endogenously increasing *Gdnf* developmentally at midgestational age E11–E12 resulted in a schizophrenia-like phenotype in adult mice, we here investigated whether methamphetamine administration during the same midgestational age may result in a similar *Gdnf* induction, potentially underlying the developmental difficulties associated with drug use during pregnancy. Although we observed an increase in *A2a* and *Drd1a* expression, we did not observe any changes in either *Gdnf* expression or the expression of its receptors *Gfra1* and *Ret*, however. Thus, methamphetamine may not induce changes to GDNF or its receptors during pregnancy, although future studies at different time points and dosages could examine this potential lack of effect in more detail.

Taken together, our results suggest that acute methamphetamine induces an acute increase in striatal *Gdnf* expression, which in turn may enhance *Ret* expression in dopamine cell bodies in the substantia nigra, as reported earlier [15,84,86]. It is well established that, due to its functional importance, the dopamine system is strictly regulated by various regulatory feedback loops [87]. Thus, it is perhaps not surprising that here we provide evidence for yet another mechanism—spatial control over *Ret* mRNA abundance in the dopamine cell bodies and axons. Since methamphetamine and GDNF both act as strong stimulants of dopamine function, and GDNF stimulates *Ret* mRNA expression, dampening striatal dopamine function via suppression of axonal *Ret* mRNA abundance in the striatum is an intuitive mechanism to reduce striatal hyperdopaminergic function. This is also well in line with our previous results demonstrating dampened dopamine function after an adult-onset 50% reduction in striatal *Gdnf* expression [13]. Further experiments to delineate molecular mechanisms through which *Ret* mRNA axonal abundance is regulated could also reveal novel drug targets for treating methamphetamine-induced psychosis in the future. Additionally, further reinforcement of this hypothesis via histological analysis of synaptic changes in nigrostriatal dopamine neurons, as well as further behavioral analysis of these animals, is an important topic of future study. A summary of our findings and the proposed mechanism of individual susceptibility to psychosis stemming from our research is summarized in Figure 6. Briefly, acute methamphetamine treatment, in addition to direct release of dopamine, induces the expression of dopamine system enhancer *GDNF*, whereas chronic methamphetamine reduces the expression of *GDNF* receptors, likely as a compensatory mechanism.

## Figures and Tables

**Figure 1 biomolecules-13-01428-f001:**
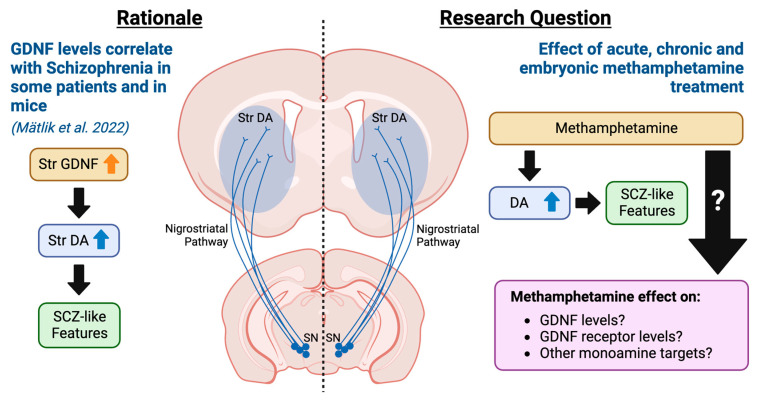
Rationale and research questions of the current exploratory study. Previously, we showed that GDNF, a potent enhancer of nigrostriatal dopamine, is elevated in a sub-set of SCZ patients and that a similar increase in GDNF in mice is sufficient to drive disease [2]. Here, we investigate how methamphetamine, known to increase the risk of FEP, modulates GDNF, its receptors, and monoamine function-related gene expression after acute, chronic, and embryonic methamphetamine exposure.

**Figure 2 biomolecules-13-01428-f002:**
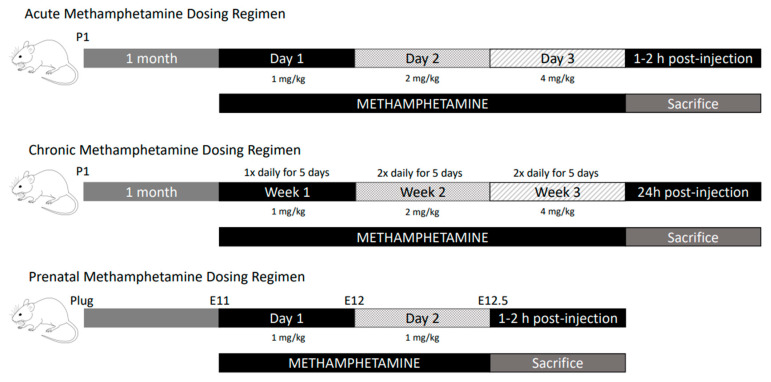
Schematic drawing of methamphetamine dosing regimens.

**Figure 3 biomolecules-13-01428-f003:**
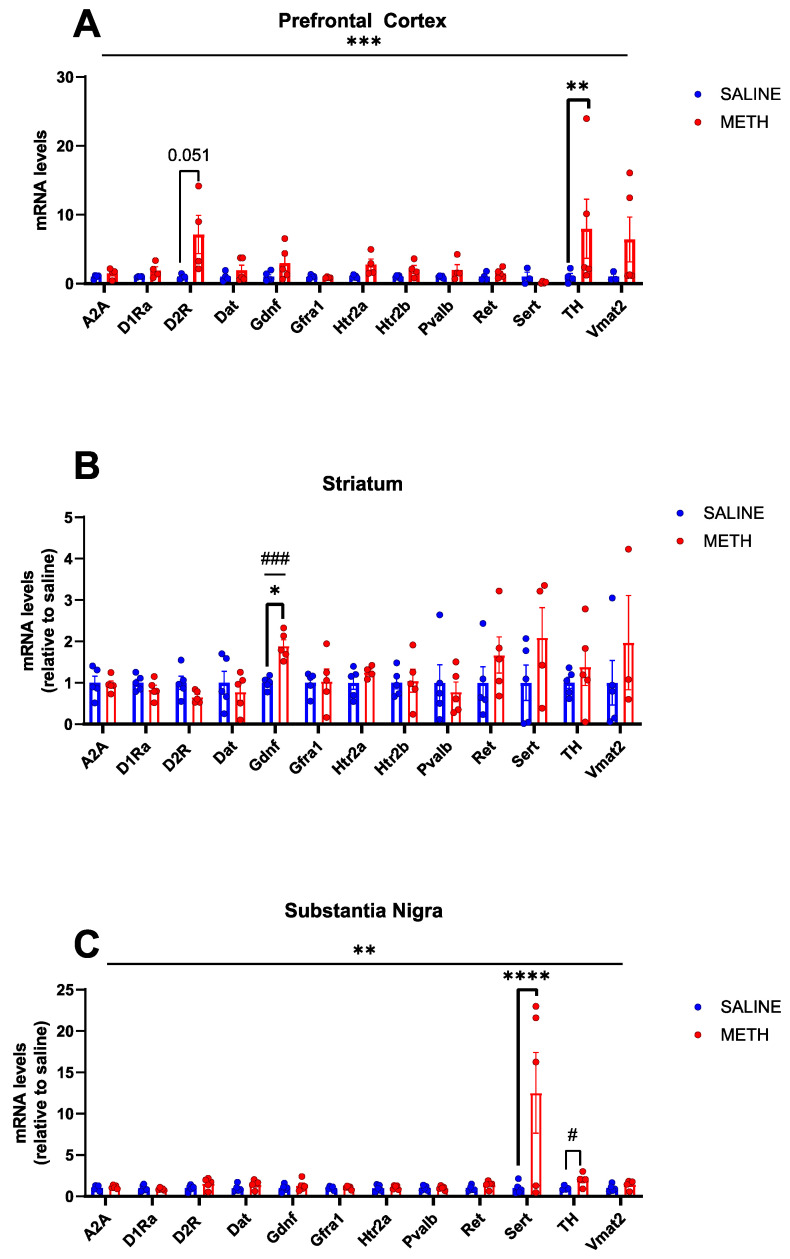
Effects of acute methamphetamine treatment on mRNA levels in WT adolescent mice, as relative to saline controls. (**A**) In the PFC, the overall effect of treatment is significant (*p* < 0.001 ***), and a significant increase in *TH* mRNA levels is observed in response to acute methamphetamine treatment (*p* < 0.01 **) and a near significant increase in *D2R* mRNA levels in response to acute methamphetamine (*p* = 0.051). (**B**) In the STR, there is a significant increase in *Gdnf* mRNA levels in response to acute methamphetamine (*p* < 0.05 *, *p* < 0.001 ###). (**C**) In the SN, there is an overall significant effect of the treatment (*p* < 0.01 **) and a significant increase in *SERT* mRNA levels in response to acute methamphetamine treatment (*p* < 0.0001 ****), and a significant increase in TH is observed when comparing saline vs. methamphetamine mRNA levels with a Student’s *t*-test (*p* < 0.05 #). N = 4–5 mice per group. * Significance measured by two-way ANOVA with Holm-Sidak post hoc test. # Significance measured by Student’s *t*-test.

**Figure 4 biomolecules-13-01428-f004:**
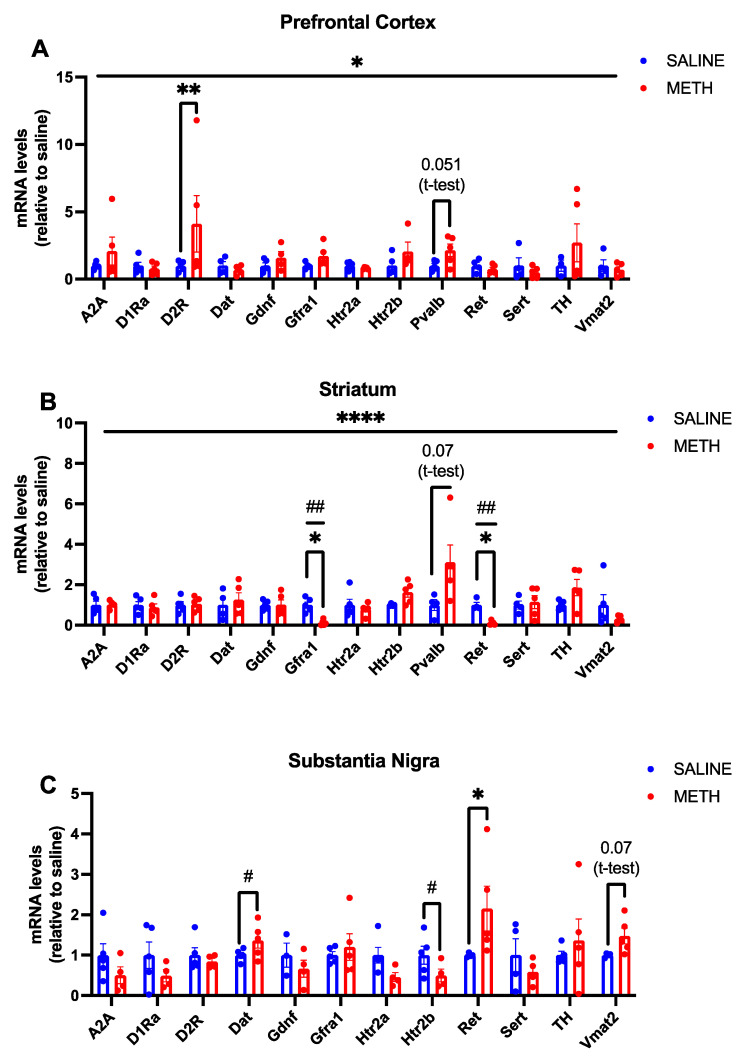
Effects of chronic methamphetamine treatment on mRNA levels in WT adolescent mice. (**A**) When results were expressed as relative to the saline control, there was a significant effect of treatment (*p* < 0.05 *) but no significant interaction was observed. Chronic methamphetamine treatment significantly increased *D2R* mRNA levels (*p* < 0.01 **), and a trend towards increased *Pvalb* mRNA levels was observed (*p* = 0.051). (**B**) In the striatum, an overall significant interaction by ANOVA between gene and treatment is observed (*p* < 0.0001 ****). A significant reduction in *Gfra1* and *Ret* mRNA levels is also observed (*p* < 0.05 *; *t*-test result p < 0.01 ##) and a trend towards an increase in *Pvalb* mRNA levels is detected in response to chronic methamphetamine treatment (*p* = 0.07). (**C**) In the SN, there is an overall significant interaction between the genes and treatment (*p* < 0.05 *) and an overall significant difference in genes (*p* < 0.05 *). A significant increase in *Dat* (*p* < 0.05 #) and *Ret* (*p* < 0.05 *) mRNA levels was observed and a trend towards increased *Vmat2* mRNA levels was found (*p* = 0.07) in response to chronic methamphetamine treatment. N = 4–5 mice per group. * Significance measured by two-way ANOVA with Holm-Sidak post hoc test. # Significance measured by Student’s *t*-test.

**Figure 5 biomolecules-13-01428-f005:**
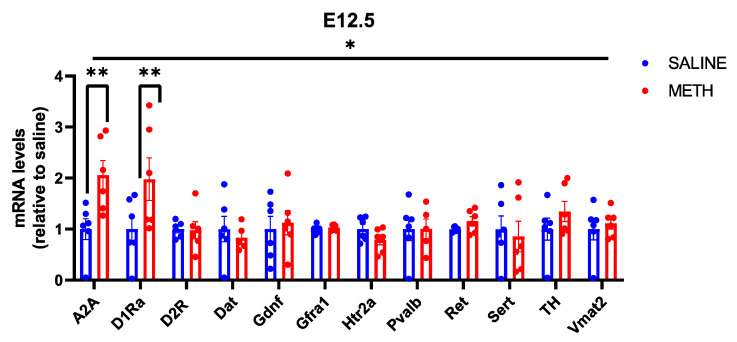
Effects of methamphetamine treatment on mRNA levels in E12.5 mice, when expressed as relative to the saline control. There is a significant effect of both gene (*p* < 0.05 *) and treatment (*p* < 0.05 *), as well as a significant interaction between gene and treatment (*p* < 0.05 *). The significant increase in A2A mRNA levels observed in response to methamphetamine was maintained (*p* < 0.01 **) and a significant increase in D1Ra was detected in response to methamphetamine treatment (*p* < 0.01 **). N = 6 mice per group. * Significance measured by two-way ANOVA with Holm-Sidak post hoc test.

**Figure 6 biomolecules-13-01428-f006:**
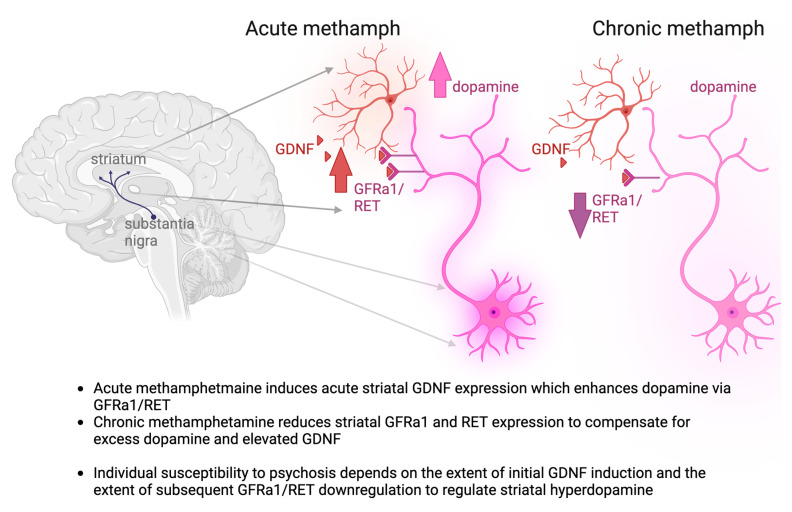
Summary of our findings and the proposed mechanism of individual susceptibility to psychosis. Red neuron—GDNF producing striatal interneuron, red hue—secreted GDNF; violet neuron—dopamine neuron of the substantia nigra, violet hue—secreted dopamine.

**Table 1 biomolecules-13-01428-t001:** Primers used for RT-qPCR.

Primer	Forward Sequence	Reverse Sequence
*adora2a*	TGG GAG CCA GAG CAA GA	GCA GCC CTT TCC TCA CAA GA
*actin*	CTA AGG CCA ACC CTG AAA AG	ACC AGA GGC ATA CAG GGA CA
*b2m*	CTC GTT GAC CCT GGT CTT TC	TTG AGG GGT TTT CTG GAT AG CA
*dat*	AACCTGTACTGGCGGCTATG	GCTGACCACGACCACTACA
*drd1a*	GCG TGG TCT CCC AGA TCG	GCA TTT CTC CTT CAA GCC CCT
*drd2*	ACA CAC CGT ACA GCT CCA AG	GGA GTA GAC GAC CAC GAA GGC AG
*gapdh*	GCC TCG TCC CGT AGA CAA AA	ATG AAG GGG TCG TTG ATG GC
*gdnf*	CGC TGA CCA GTG ACT CCA ATA TGC	TGC CGC TTG TTT ATC TGG TGA CC
*gfra1*	TTC CCA CAC ACG TTT TAC CA	GCC CGA TAC ATT GGA TTT CA
*htr2a*	AAC CCC ATT CAC CAT AGC CG	CCG AAG ACT GGG ATT GGC AT
*htr2b*	TGC CCT CTT GAC AAT CAT GT	AGG GAA ATG GCA CAG AGA TG
*pvalb*	TGG AGA CAA GGA TGG GGA CG	CCA CTT ACG TTT CAG CCA CC
*ret*	TCC CTT CCA CAT GGA TTG A	ATC GGC TCT CGT GAG TGG TA
*rps6*	GGT TGG GAC CTA AAA GGG CT	GGT CCT GGG CTT CTT ACC TT
*slc6a4*	CCC AGA CTC TTG TGG GTT CC	CTA GCT GAT GAC TGG GTG GC
*th*	CCCAAGGGCTTCAGAAGAG	GGGCATCCTCGATGAGACT
*vmat2*	ATGCTGCTCACCGTCGTAGT	TTTTTCTCGTGCTTAATGCTGT

## Data Availability

All data related to this manuscript are available upon request.

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
