# Peer review of "Analysis of Acute and Chronic Methamphetamine Treatment in Mice on Gdnf System Expression Reveals a Potential Mechanism of Schizophrenia Susceptibility"

_biomolecules, 2023, doi:10.3390/biom13091428_

Round 1
Reviewer 1 Report (New Reviewer)
This study investigated the effects of acute and chronic methamphetamine treatment in mice on GDNF system. The authors found that acute methamphetamine application increases Gdnf expression in the striatum and chronic methamphetamine decreases the striatal expression of GDNF receptors Gfra1 and Ret. They concluded that these alterations were related to susceptibility to schizophrenia. Overall, this study addresses am important psychiatric issue. However, it is required some corrections.
1) The authors concluded that methamphetamine-induced the alteration of GDNF system was related to the susceptibility to schizophrenia. However, the authors examined the mice a few hours or 24 hours after administering methamphetamine to the mice. In addition, they did not evaluated behavior of the mice. Therefore, the present study did not demonstrate susceptibility to schizophrenia, but only the effect of methamphetamine on the GDNF system.
2) It is not clear why the experiment using pregnant mice was conducted. In addition, they did not mention this result in the discussion.
3) It is necessary to consider whether there is any discrepancy between mRNA and protein expression levels in the receptors.
This manuscript needs a language check.
Author Response
Please see the attachment

Reviewer 2 Report (New Reviewer)
In general, I think the subject of article by Casserly and co-authors is really interesting, and the authors’ fascinating observations on this timely topic may be of interest to the readers of Biomolecules. However, some comments needed to be addressed to improve the quality of the manuscript prior to its publication.
Major comments:
1) Why did the authors choose a two-way ANOVA although they have animals of one genotype and only one variable parameter – drug administration? It is not clear from the description of statistical methods what was compared with what. Actually in this case it would be correct to use only t-test. Whether the samples were tested for normal distribution? Given that the number of animals is small, the probability that the distribution in some cases may differ from the normal increases, which means that a non-parametric method of comparison is required. Looking at the graphs, it seems that the distribution of values, for example, for the TH mRNA level in Figure 3A or Pvalb in Figure 4B may differ from normal and requires the use of the Mann-Whitney test. Also, the adequate description of statistical data is absent throughout the “Results” section, t-test values and degrees of freedom are not indicated. Thus, authors should provide a rationale for used statistical methods as well as adequately describe the results obtained.
2) In the absence of data on Ret phosphorylation the authors should tone down their speculations on drug targeting (lines 536-543). Moreover GFRa1 and Ret transcriptionally are not responded to acute methamphetamine application and their transcripts number even decreased in projection areas after chronic methamphetamine treatment. Suggestion on the redistribution of Ret along DA neurons (lines 544-578) is still discursive. So, the idea of blocking the GDNF signaling is not so obvious.
3) Authors should explain why the mixed genetic background 129OIa/ICR/C57bl6 was used.
4) It is need to be clarify why a pregnant dams were injected with methamphetamine for two days and why exactly at E11-E12?
Minor comments:
1) For a better visual perception of graphs, authors need to make them a little larger.
2) Throughout the text, when the authors talk about the expression of Gdnf they mean the gene, hence the abbreviation should be written in italics. This also applies to any other abbreviations of genes that appear in the text of the manuscript.
Round 2
Reviewer 1 Report (New Reviewer)
Thank you for your sincere response to my points.
The study addresses an important psychiatric issue and the paper is well-written.
Reviewer 2 Report (New Reviewer)
The authors have adequately addressed all comments. I have no further concerns regarding manuscript content
This manuscript is a resubmission of an earlier submission. The following is a list of the peer review reports and author responses from that submission.
Round 1
Reviewer 1 Report
Paper titled (Analysis of acute and chronic methamphetamine treatment in mice on Gdnf system expression reveals a potential mechanism of schizophrenia susceptibility) by Cassereli et al. is NOT a solid novel idea. This study was better amended by some behavioral tasks to explore psychotic behaviors. Methods is done collectively and needs organization & subtitling. No drug control groups were assigned in this stduy.
I have the following comments:
1- Title: write GDNF not gdnf
is not perfect in language I suggest to be : Impact of acute and chronic methamphetamine treatment in mice on Gdnf system expression reveals a potential mechanism of schizophrenia susceptibility
Also the formulation is strange: does only the effect of amphetamine on GDNF is the cause that of induction of schizophrenia??!!
2- Abstract is not fine and should contain some numerical values.
3- Intro: the first paragraph about Schizophrenia is too long: please make it concise & give more details for amphetamine mechanism and why it would induce schizophrenia (3rd paragraph)
Also the aim should be refined and written clear & how you achieved it.
what the authors wrote is a conclusion NOT an aim "Our results may eventually provide a rationale for subtype specific treatment strategies 87 based on GDNF pathway inhibition and encourage future research on GDNF levels, dis- 88 ease outcome and patient stratification. "
4- What was the age, weight & sex of the animals?
5- Authors should give the source of chemicals, kits and antibodies completely and consistently (code, company, town, state and country) & version for software
6- In figure 1: write the dose of acute & cheonic amphetamine
7- What was the equation used for calaculation of PCR results? write the standard reference. should be written in PCR methods
8- Unsing student t test is NOT appropriate in this study design
Authors have to check the normality of distribution of the results by a suitable post hoc test (such as Shapiro-Wilk test or K-S test) before deciding to choose certain ANOVA. If the normality test indicated normal dist of the data, so use one-way ANOVA, if not, use non parametric ANOVA
In all cases choose a suitable post-hoc test
9- In methods: write and confirm that you highlighted every possible comparison between the study groups.
10- Why authors have chosen to study SN, striatum for a schizophrenia model? did author think about limbic system ?
11- No need for P values <0.01, 0.001 , this does not mean your means are very different!, just means that SD values are small.
12- How euthanasia was performed and brain was dissected?
13- English use should be revised by a native speaker or a colleague proficient in English
14- Give the method of solubilizing or dilution of the drugs.
Reviewer 2 Report
It is interesting, well written manusript. It contains description of acute and chronic methamphetamine treatment in mice, comparison of this two ways of treatment and observed effects correlation with schizophrenia signs and symptoms. Influence on GDNF system expression was discussed. Such research may have influence on future schizophrenia treatment considering new mechanism of therapeutic intervention. To enhance quality of results presentation I propose to include in manuscript List of Abbreviations and to try to avoid too frequent abbreviations use. It will be more friendly for readers
Reviewer 3 Report
The manuscript by Casserly et al., describes the effect of acute and chronic methamphetamine treatment on GDNF system expression in adult and young mice, suggesting a potential mechanism of schizophrenia susceptibility.
The topic of the study is not extremely original and the experimental design still preliminar as well as the results described . Further experiments have to be performed in order to explain the differences observed in GDNF expression, in acute and chronic conditions, in the different brain regions examined. Moreover, the consequences of GDNF expression on dopaminergic activity in mice exposed to chronic and acute methamphetamine treatment should be explored in order to validate the working hypothesis.
Major points:
1. The differences in the increase in mRNA for GDNF and its signalling related pathway (GFRa1 and Ret) in chronic, acute methamphetamine treatment in adult and embryonic mice should be supported by behavioural experiments as well as by biochemical analysis of the potential mechanistic differences between acute and chronic treatment in adult and embryonic mice. These experiments would reinforce the hypothesis that Gdnf induction and subsequent Gfra1 and Ret downregulation may be responsible for the individual susceptibility to psychosis. Moreover, these experiments would also provide the proof of concept for future precision medicine development aimed to using GDNF/GFRa1/RET antagonists for methamphetamine induced psychosis.
2. The scheme reported in figure 5 summarizing the results obtained in the animal model needs to be implemented because it is not clear in the present form.
3. Further experimental approaches should be performed in order to demonstrate the changes in synaptic plasticity occurring in substantia nigra of the mice exposed to acute or chronic treatments with methamphetamine (number and shape of dendritic spines).